# Muscle Enriched Lamin Interacting Protein (*Mlip*) Binds Chromatin and Is Required for Myoblast Differentiation

**DOI:** 10.3390/cells10030615

**Published:** 2021-03-10

**Authors:** Elmira Ahmady, Alexandre Blais, Patrick G. Burgon

**Affiliations:** 1Department of Biochemistry, Microbiology, and Immunology, University of Ottawa, Ottawa, ON K1H 8M5, Canada; elmiraahmady25@gmail.com (E.A.); alexandre.blais@uottawa.ca (A.B.); 2Molecular Signaling Laboratory, University of Ottawa Heart Institute, Ottawa, ON K1Y 4W7, Canada; 3Ottawa Institute of Systems Biology, University of Ottawa, Ottawa, ON K1H 8M5, Canada; 4University of Ottawa Centre for Infection, Immunity and Inflammation (CI3), Ottawa, ON K1H 8M5, Canada; 5Department of Chemistry and Earth Sciences, College of Arts and Sciences, Qatar University, Doha 999043, Qatar

**Keywords:** *Mlip*, ChIP, skeletal muscle

## Abstract

Muscle-enriched A-type lamin-interacting protein (*Mlip*) is a recently discovered Amniota gene that encodes proteins of unknown biological function. Here we report *Mlip*’s direct interaction with chromatin, and it may function as a transcriptional co-factor. Chromatin immunoprecipitations with microarray analysis demonstrated a propensity for *Mlip* to associate with genomic regions in close proximity to genes that control tissue-specific differentiation. Gel mobility shift assays confirmed that *Mlip* protein complexes with genomic DNA. Blocking *Mlip* expression in C2C12 myoblasts down-regulates myogenic regulatory factors (MyoD and MyoG) and subsequently significantly inhibits myogenic differentiation and the formation of myotubes. Collectively our data demonstrate that *Mlip* is required for C2C12 myoblast differentiation into myotubes. *Mlip* may exert this role as a transcriptional regulator of a myogenic program that is unique to amniotes.

## 1. Introduction

A-type lamins have been implicated in the maintenance of cellular commitment and differentiation. Mutations in the A-type lamin gene (*LMNA*) manifest as tissue-specific degenerative diseases (laminopathies) with varied clinical symptoms that include dilated cardiomyopathy, atherosclerosis, muscular dystrophy, lipodystrophy, neuropathy and progeria [1,2,3,4]. The molecular mechanisms that link mutations in *LMNA* with different human diseases are poorly understood at this time. The recent discovery of a unique muscle-enriched A-type lamin-interacting protein (*Mlip*) [5] may provide valuable new insight into how primary genetic defects in *LMNA*, a ubiquitously expressed nuclear protein, translates to tissue-specific diseases.

*Mlip* is expressed ubiquitously and most abundantly in heart, skeletal and smooth muscle. The *Mlip* gene encodes at least seven, alternatively spliced, A-type lamin-interacting factors that possess a unique primary amino-acid sequence not found in other proteins [5,6]. *Mlip* is observed only in amniotes’ genomes as a single copy gene with no evidence of duplication [5], suggesting that there may be no functional redundancy. In addition, the type of *Mlip* isoforms expressed differs between each of the tissues, with the heart being the most heterogeneous in *Mlip* isoform expression [5,6]. Down-regulation of lamin A/C expression by shRNA results in *Mlip*’s up-regulation and its mislocalization from the nuclear envelope to the nucleoplasm/cytoplasm [5]. Recently, *Mlip* was shown to interact with Islet1 (Isl1), an essential transcription factor for cardiac progenitor cell specification [7,8], and blocks agonist-induced hypertrophy of cardiomyocytes in culture [9]; *Mlip* plays a critical role in the maintenance of cardiac homeostasis and protects the heart against pathophysiological stresses [10,11] and for normal myonuclear positioning in skeletal muscle [6,12]. *Mlip*’s precise biological role remains to be defined.

Here we report the functional characterization of *Mlip* as a chromatin-binding protein with transcriptional co-factor activity. *Mlip* interacts with chromatin in close proximity to many genes that are functionally associated with tissue development and required for the normal myoblast fusion process during differentiation. The findings presented in the current study provide the first insight into the biological function of *Mlip*.

## 2. Materials and Methods

### 2.1. Cell Culture

The mouse C2C12 myoblast cell line (American Type culture collection ATCC no. CRL-1772) was cultured at 37 °C in an atmosphere of 10% CO_2_ in growth medium consisting of Dulbecco’s modified Eagle’s medium (DMEM) supplemented with 20% fetal bovine serum (FBS), 1% *v/v* penicillin-streptomycin and 2 mM L-glutamine. To induce myogenic differentiation, cells with 60–70% confluency were subjected to differentiation by switching growth medium to differentiation medium (DMEM + 2% horse serum). For immunofluorescence, staining cells were grown on sterile coverslips coated with gelatin in 12-well cell culture treated dishes.

The mouse HL1 cell line was cultured at 37 °C in an atmosphere of 5% CO_2_, in Claycomb medium (Sigma, Saint louis, MO, USA) supplemented with 10% fetal bovine serum, 1% *v/v* penicillin-streptomycin and 2 mM L-glutamine. The HEK293 cell line was cultured at 37 °C in an atmosphere of 5% CO_2_, in DMEM medium (Thermofisher, Waltham, MA, USA), supplemented with 10% fetal bovine serum, 1% *v/v* penicillin-streptomycin and 2 mM L-glutamine.

### 2.2. Whole Cell Extract Preparation for Immunoprecipitation and Western Blotting

Proteins were extracted using lysis buffer containing: 50 mM Tris-HCl [pH = 8.0], 200 mM NaCl, 20 mM NaF, 20 mM β-glycerolphosphate, 0.5% NP-40, 0.1 mM Na_3_Vo_4_, 1 mM dithiothreitol, 1 × Protease inhibitor cocktail (Roche, 1 tablet/7.0 mL), and phosphatase inhibitor cocktails (Sigma, 0.1 mL/7.0 mL). Cells were scraped from dishes and lysates pipetted into 1.5 mL centrifuge tubes. Lysates were incubated on ice for 20 min, and cleared by centrifugation at 10,000× *g* for 10 min at 4 °C. Supernatants collected and protein concentrations were determined using the Bradford protein assay (Biorad). Proteins were resolved on a 15% 0.75 mm thick SDS-TrisGlycine-Polyacrylamide gel (200 V) and transferred to PVDF membranes (Millipore) for 1 h at 100 V at 4 °C. Membranes were blocked in 5% nonfat milk dissolved in Tris-buffered saline +0.05% Tween 20. *Mlip* mediated immunoprecipitations were performed as described previously [5].

For Western blot analysis, *Mlip* antibodies were raised in rabbits against a synthetic peptide N-LRKDEEVYEPNPFSKYL-C (21st Century Biochemicals) [5]. The following primary antibody concentrations were used: *Mlip*, 1:50,000 of rabbit antibody, Myogenin mouse monoclonal IgG 1:1000 (Clone F5D sc12732; Santa Cruz Biotechnology, Dallas, TX, USA) and anti-alpha-Tubulin mouse monoclonal 1:5000 (T9026, Sigma). MyoD mouse monoclonal IgG 1:500 (Clone 5.8A, sc32758; Santa Cruz Biotechnology) as well as Six1 and Six4 rabbit polyclonal 1:1000 [13].

Membranes were then incubated with HRP-conjugated anti-rabbit or anti-mouse IgG secondary antibody (Santa Cruz Biotechnology). Immunoblot signals were detected with a SuperSignal West Pico Chemiluminescent Kit (Thermo Scientific, Waltham, MA, USA) and visualized on X-ray film (Sigma).

### 2.3. Immunofluorescence

C2C12 cells were grown on gelatin-coated coverslips in 12-well cell culture treated plates (Fisher Scientific, Waltham, MA, USA). C2C12 cells were differentiated using differentiation medium and fixed at appropriate time points using pre-cooled methanol for 20 min on ice and were washed carefully twice with phosphate-buffered saline (PBS). To reduce nonspecific binding, cells were blocked with PBS supplemented with 5% fetal bovine serum and 0.1% NP-40 in PBS for 30 min at room temperature with gentle shaking and were then incubated with anti-*Mlip* (1:1000), anti-MHC antibody (1:500) (Developmental Studies Hybridoma Bank) in 1.5% FBS/PBS for 1 h at room temperature followed by fluorescent-conjugated secondary antibody Regulus Red 594 anti-mouse IgG (LP Bio). Nuclei were stained with DAPI (1 µg/mL) and mounted with fluorescent mounting medium (Dako). Images were then visualized using epifluorescent microscopy (Carl Zeiss Axio Imager 2) using objective magnification: 10×, numerical aperture: 0.25, Plan-Apochromat or 20 X, numerical aperture: 0.8, Plan-Apochromat. The Illuminator HBO 100 for fluorescence applications was used as a light source. The following Zeiss filter sets were available (46HE; 38HE; 43HE; 47HE; and 25HE) and images were captured with a ZEISS Axiocam 506 color and Axio cam HRM using the Zeiss Zen 2.3 software.

### 2.4. Chromatin Immunoprecipitation and ChIP-on-Chip Studies

C_2_C_12_ myoblast cells were grown in 5 × 150 mm cell culture dishes (50 × 10^6^ cells) and subjected to chromatin immunoprecipitation according to manufacturer′s protocol (cell signaling). Chromatin was sheared into approximately 5 nucleosomes. Chromatin (250 ug) was immunoprecipitated with a specific *Mlip* antibody (1:1000), Histone H3 (positive control) and normal rabbit IgG (negative control). DNA linker’s oJW102: 5′-GCG GTG ACC CGG GAG ATC TGA ATT C-3′ and oJW103: 5′-GAA TTC AGA TC-3′ were ligated to recover *Mlip* ChIP DNA according to the method described by Agilent Mammalian ChIP-on-chip protocol (Agilent). DNA amplification was carried out as follows: 95 °C for 5 min, followed by 34 cycles in 3 steps: 95 °C for 30 s, 60 °C for 30 s, and 72 °C for 30 s and 72 °C for 5 min using Taq DNA polymerase (NEB). PCR products were cleaned using QiAquik PCR purification kit columns (Qiagen). For ChIP DNA sequencing, 4.0 µL of *Mlip* linkered PCR product was TOPO-TA cloned into pCRII-TOPO (Invitrogen) and transformed into bacteria.

ChIP-on-chip was performed according to Agilent Mammalian ChIP-on-chip protocol. Putative *Mlip* targets were analyzed using the UCSC Cis-Genome Browser database and DAVID (Database for Annotation, Visualization and Integrated Discovery, 2008) bioinformatics tool for analysis of molecular function ontologies. Gene-specific ChIP-PCR was performed with the following primers RARA (5′-TTC TTT CCC CCT ATG CTG GGT-3′; 5′-GGG AGG GCT GGG TAC TAT CTC-3′), SIX1 (5′-ATG CTG CCG TCG TTT GGT T-3′; 5′-CCT TGA GCA CGC TCT CGT T-3′), SIX3 (5′-CCG GAA GAG TTG TCC ATG TTC-3′; 5′-CGA CTC GTG TTT GTT GAT GGC-3′), SIX4 (5′-CCA CGG TTT TTC CCT GAC CC-3′; 5′-GGT TGC ATA GTT AGT GTT GCT GA-3′) and MyoG (5′-GCG GAC TGA GCT CAG CTT AAG-3′; 5′-GCT GTC CAC GAT GGA CGT AAG-3′).

### 2.5. Nuclear Extracts and Electrophoretic Mobility Shift Assays (EMSA)

Nuclear protein extracts from cultured C2C12 myoblasts were prepared essentially as described by Farrance and Ordahl [14]. DNA-protein binding was assayed with a double stranded DNA oligonucleotide probe that was biotin 3′end DNA labeled (Pierce) or ^32^ P-labelled. EMSA were performed as described by Ueyama et al. [15]. Reactions were performed in binding buffer (20 mM HEPES (pH 7.9), 10% glycerol, 50 mM KCl, 0.05% NP-40, 0.5 mM EDTA, 0.5 mM DTT, and 1 mM PMSF) in the presence of 0.5 µg of Poly (dI-dC), a nonspecific competitor; *Mlip* protein generated by in vitro coupled transcription-translation using the TNT Coupled T7/Sp6 Wheat Germ extract system (Promega, Madison, WI, USA) for 20 min at room temperature. Products of the binding reactions were resolved by 6%polyacrylamide gel electrophoresis (PAGE) 0.5 X TBE gel for 3–4 h at 10 mM. Biotin labeled probe binding reactions were transferred to a nylon membrane (Millipore, Burlington, MA, USA) and biotin-labeled DNA was detected by chemiluminescence (Promega). Then 32 P-labelled probe gels were dried and analyzed with a phosphorimager (Amersham). DNA oligonucleotide probe sequence for EMSA, the sequence for Six3 (5′-GCA GGA TCC CTA CCC CAA CCC CAG CAA GAA ACG C-3′; 5′-GCG TTT CTT GCT GGG GTT GGG GTA GGG ATC CTG C-3′).

### 2.6. Reporter Assay

Luciferase reporter assays were performed with the Mammalian Two-Hybrid Assay Kit (Agilent Technologies). *Mlip* was subcloned in frame with a GAL4-DBD of the pCMV-BD vector and subsequently co-transfected into HEK293 cells with a pFR-Luc reporter plasmid that contained the GAL4 promotor driven Luciferase using Lipofectamine reagent (Invitrogen). Twenty-four h post-transfection lysates were harvested with a passive lysis buffer (Promega) and luciferase activity was determined by a Dual-Luciferase reporter assay system (Promega) as per the manufacturer’s protocol.

### 2.7. Transfection and Generation of Stable Mlip Knockdown C2C12 Cell Lines

Two mouse *Mlip*-shRNAmir sequences in the pGIPZ vector were purchased from Open Biosystems and non-silencing shRNAmir Control vector (Dharmacon RHS4346). Sequences included: V2LMM-214053 mature-sense: CCA ACT ACT TGC TAA ACT T (mm39 Chr9: 77009596-77009616) and V2LMM-211301 mature-sense: CCT ATA ATG CCT TCT ATT A (mm39 Chr9: 77009435-77009454). C2C12 cells in 60 mm plates were transfected with 4.0 µg of shRNAmir plasmid DNA or non-silencing shRNAmir Control vector using Arrest-In transfection (Open Biosystems) reagent in serum-free, antibiotic-free media. Control cells were generated using a validated non-silencing negative control expressed in the pGIPZ vector. Stable cell lines were established using puromycin (1.5 µg/mL) resistance selection.

## 3. Results

### 3.1. Mlip Interacts with Chromatin in Areas of Close Proximity to Developmental Genes

We recently reported the discovery of *Mlip* through its interaction with lamin A/C [5]. In addition to interacting with lamin A/C, we also demonstrated that *Mlip* co-localizes with the promyelocytic leukemia protein (PML) a tumor suppressor protein. Previous studies demonstrated that lamin A/C and many of the proteins within PML bodies formed microdomains of chromatin organization and transcription [16,17]. *Mlip* also localizes to micro-domains in the nucleus in close proximity to chromatin, in undifferentiated mouse C2C12 myoblasts (Figure 1A). To address if *Mlip* also interacts with chromatin, we performed chromatin immunoprecipitation (ChIP) assays. Since the heart expresses the largest number of *Mlip* splice forms, we first determined that our *Mlip* antibody was effective in immunoprecipitating endogenous *Mlip* from total heart lysates (Figure 1B). We then determined that chromatin was immunoprecipitated through *Mlip*, in four independent ChIP experiments, using the *Mlip* specific antibody against chemically crossed-linked *Mlip* of undifferentiated C2C12 myoblasts. A smear of *Mlip*-bound DNA (Figure 1C) was observed.

To gain insight into the DNA sequence immunoprecipitated through *Mlip*, the *Mlip* enriched chromatin was initially TA-cloned [18], followed by direct-PCR of clones and subsequent sequencing (Figure 1D). Ontological analysis of the genes in close proximity to the *Mlip*-ChIP fragments was found to be primarily associated with commitment, differentiation, apoptosis and growth (Table 1). With this data, we identified that mouse chromosome 2 (182 Mbp) and 11 (122 Mbp) enrich for similar gene ontologies.

To further identify *Mlip* chromatin targets, we designed DNA arrays that tiled out the smaller mouse chromosome 11 (1690 genes, Appendix A), thereby allowing for an additional 1600 promoter regions (−5 kbp to + 2 kbp around the transcriptional start sites) of genes on the remaining mouse chromosomes (Appendix A) with similar ontologies as those observed in our preliminary studies (Table 1) to be tiled out on the DNA arrays. The *Mlip* ChIP-enriched DNA was labeled and hybridized to our custom designed Agilent DNA microarrays (ChIP-on-Chip). The results from two independent biological replicates identified 133 putative *Mlip* chromatin targets (Appendix A).

To minimize bias, since only mouse chromosome 11 was tiled on the DNA array, analysis of the ontological enrichment of the 45 putative *Mlip* gene targets identified on chromosome 11 (Appendix A) by DAVID Bioinformatics Resources [19,20] revealed a significant enrichment of genes that are involved in multicellular organismal development (*p* < 0.000044) and developmental process (*p* < 0.001) (Table 2) as compared to all chromosome 11 encoded genes. Gene-specific ChIP-PCR was performed on a selected number of identified *Mlip* targets to verify our ChIP-on-Chip data (Figure 2A). We observed myogenin (MyoG), retinoic acid receptor alpha (RARα) and Sine oculis homeobox homolog 1/4 (Six1/4) in the *Mlip* mediated immunoprecipitated chromatin (Figure 2A).

We performed Electrophoretic Mobility Shift Assays (EMSA) on targets identified by the ChIP-on-Chip assay (Appendix A) to substantiate these results. A Gel Super-Shift Assay with our *Mlip* specific antibody showed that *Mlip* is part of the Six3 shifted complex (Figure 2B). This super-shifted complex was disrupted by specific competition with unlabeled Six3 competitors as well as with a peptide neutralized *Mlip* antibody. Results demonstrated that the mobility shift was specifically due to *Mlip*’s association with the DNA–protein complex (Figure 2B). To determine if the binding of *Mlip* with the SIX3 probe was direct, *Mlip* was translated in vitro using a cell-free, wheat-germ-based, transcription-coupled translation system. The biotin-labeled SIX3 probe was incubated with the in vitro translated *Mlip*. We observed a direct interaction between *Mlip* and the Six3 sequence (Figure 2C). A 50-molar excess of unlabeled Six3 probe competed out the labeled Six3 probe. This strongly suggests that *Mlip* is able to interact directly with DNA.

### 3.2. Mlip Contains a Transcriptional Activation Domain

To determine if *Mlip* contains a transcriptional activation domain, we performed co-transfection experiments with HEK293 cells using a GAL4 promoter-luciferase reporter construct together with an expression vector in which the Gal4-DNA-binding-domain was fused to *Mlip* (GAL4-DBD-*Mlip*). We found that GAL4-DBD-*Mlip* significantly (*p* < 0.01) trans-activated the luciferase reporter through the GAL4 promoter by ~8-fold over the control in HEK293 cells (Figure 2D). Expression of the N-terminal half of *Mlip* (exons 1–3) fused to a GAL4-DBD (GAL4-DBD-*Mlip*-N) resulted in a significant (35%, *p* < 0.05) reduction in the transcriptional activity of *Mlip*, whereas the expression of the C-terminal half of *Mlip* (exons 5–12, GAL4-DBD-*Mlip*-C) led to a significant (25%) increase in *Mlip*’s transcriptional activity (Figure 2D). *Mlip*’s intrinsic transcriptional activity is associated with both the N- and C-terminal regions of *Mlip*, as both activate transcription, with the C-terminal region of *Mlip* being the more potent of the two within the context of GAL4 fusion (Figure 2D). The C-terminal region of *Mlip* is comprised of the highly conserved *Mlip* domain that encompasses exons 8 to 12 [5]. Taken together, our data strongly supports the view that *Mlip* may function as a transcription regulator since *Mlip* is found in a complex with chromatin and is able to stimulate DNA transcription.

### 3.3. Impaired Myotube Formation in MLIP-Depleted C2C12 Myoblasts

To verify the biological significance of *Mlip*’s interaction with chromatin of C2C12 myoblasts, *Mlip* was specifically and stably knocked down in C2C12 cells utilizing shRNAmir technology [21,22]. Initiating differentiation of C2C12 myoblasts by the withdrawal of serum led to the appearance of two additional splice forms of *Mlip* (30 kDa and 45 kDa) within 0.25 days with no observed changes in the expression of the 25 kDa *Mlip* splice variant (data not shown). Total *Mlip* protein expression peaked at 2 days of differentiation and all three *Mlip* forms persisted throughout differentiation of the C2C12 myotubes (Figure 3, top left panel). The distribution of *Mlip* cellular localization during C2C12 differentiation was assessed by indirect immunofluorescence (Appendix A). In undifferentiated C2C12 myoblast (0 days) we observed punctae nuclear staining that persists at both days 1 and 2 differentiation timepoints. Myofibres (days 2 to 5) have reduced *Mlip* puntate in the nucleus with increased cytoplasmic staining (Appendix A). Taken together, these observations of a change in *Mlip* isoform expression patterns and cellular distribution suggest *Mlip* may have alternative functions in myoblasts when compared to myofibres.

Two independent *Mlip*-knockdown (KD) lines in C2C12 were established (Figure 3, top panel); KD1 and KD2 using shRNAmir’s targeted to the 3′-UTR of *Mlip*. We were able to suppress *Mlip* protein by >95% in two independent, undifferentiated C2C12 myoblast *Mlip*-knockdown lines *Mlip*-KD1 and *Mlip*-KD2; (Figure 3, top panel). Morphologically these stable, *Mlip* knocked down undifferentiated C2C12, myoblasts appeared normal. We observed no difference in the rate of cellular proliferation (doubling time) between controls (10.6 ± 1.4 h) and stably knockdown *Mlip* cells (10.3 ± 0.83 h, *n* = 3, *p* = 0.77) and these were similar to previously reported C2C12 doubling times of 12 h [23].

Upon differentiation of *Mlip*-KD1 and *Mlip*-KD2, we observed a small increase in *Mlip* expression in both knockdown lines (Figure 3) when compared to the undifferentiated myoblast *Mlip* KD lines (Figure 3, top panels). There was a significant reduction in the expression of the myogenic regulatory factors MyoD and MyoG, as compared to the parental cell line (Figure 3). The decrease in MyoD and MyoG expression was accompanied by a discernible reduction of myosin heavy chain expression in both *Mlip* knockdown lines (Figure 4A) with few multinucleated myotubes observed after 5 days of differentiation (Figure 4A, bottom row). The fusion index, the proportion of total nuclei inside myotubes, was determined after 3 days of differentiation and showed a significant (*p* < 0.001, *n* = 3) decrease in myoblast fusion between the control (35.1 ± 4%) and both *Mlip*-KD1 (2.76 ± 0.52%) and *Mlip* KD2 (10.0 ± 1.35%) (Figure 4B).

The *Mlip* ChIP-on-Chip experiments (Appendix A) identified several members of the Six homeobox gene family (Six1, Six2, Six3, Six4, and Six5) as potential targets of *Mlip*. The Six family of transcription factors, specifically Six1 and Six4, are essential for skeletal muscle development and for myoblast differentiation [13,24,25,26,27]. The knockdown of *Mlip* resulted in a significant decrease in Six1/4 expression in the undifferentiated myoblasts (Figure 3). In contrast, differentiation of the *Mlip* knockdown lines resulted in a restoration of Six1 and Six4 protein levels in a fashion similar to the control cells (Figure 3). Despite the reestablishment of Six1 and Six4 levels during differentiation, both MyoD and MyoG expression were markedly reduced and the cells lacked the ability to form normal myotubes (Figure 4). Collectively our data demonstrate that *Mlip* is required for C2C12 myoblast differentiation into myotubes and that *Mlip* may exert this role as a transcriptional regulator of the myogenic program.

## 4. Discussion

*Mlip* was originally discovered through its direct interaction with A-type lamins and is expressed in many different tissues but was found to be most abundant in striated muscle [5]. A large proportion of the mutations described in laminopathies translate into striated muscular dystrophies. We, therefore, used mouse C2C12 myoblasts derived from satellite cells, an established in vitro myoblast model system [28,29,30], to investigate the role of *Mlip* in muscle differentiation and its underlying molecular mechanism. Here we demonstrate that *Mlip* may function as a transcriptional co-factor that targets the regulation of genes involved in cell fate during development, specifically in muscle. We further establish that *Mlip* is required for muscle differentiation and may regulate the myogenic program through its association with chromatin.

Previous studies have reported that A-type lamins interact with and organize chromatin [31] that may be through a lamin-histone interaction [32]. *Mlip* is localized to both the nuclear envelope as well as within nuclear bodies, suggesting to us that *Mlip* may also interact with chromatin. Using *Mlip* mediated chromatin immunoprecipitation microarrays, we discovered that *Mlip* interacts with chromatin in close proximity to genes that encode a variety of transcription factors, many of which have functions that are critical to development and growth of skeletal muscle (PAX3, SIX1, SIX4, JUNB, and MEF2C). A-type lamins have been previously shown to interact directly with transcription factors such as pRB [33,34] SREBP-1 [35] and MOK2 [36], implying that A-type lamins could act as accessory proteins for a subset of tissue determining factors. Our data show that *Mlip*, with its tissue-specific splicing, may act as a modifier, thus providing a rationale for the phenotypic heterogeneity observed in laminopathies. Furthermore, the recent report by Huang et al. [9] demonstrated that *Mlip* also interacts with Islet1 (Isl1), an essential transcription factor for cardiac progenitor cell specification, and represses the transcriptional activity of Isl1 [37]. This raises many new and intriguing questions as to what the functional role of *Mlip* may be, especially since A-type lamins are expressed during cells/tissues commitment [38]. Both Isl1 and A-type lamins play critical roles in the regulation of gene expression and *Mlip*, through its interaction with both Isl1 and A-type lamins, may play a regulatory role for both Isl1 and A-type lamin function.

Duchenne muscular dystrophy (DMD) is characterized by progressive muscle-wasting disease that affects striated muscles including the limb muscles, the diaphragm, and the heart [39]—all tissues in which *Mlip* is expressed. Patients afflicted with DMD are diagnosed early in childhood with death usually occurring in the teenage years or early 20s by cardiorespiratory failure. Like the laminopathies, the underlying molecular pathways leading to DMD are poorly understood even though the genetic mutations (not associated with LMNA) are well characterized, and the histological pathology is well documented. Gene expression profiling of quadricep skeletal muscle biopsies from DMD patients and unaffected control patients were examined [40] and revealed a significant (*p* < 0.0001, *n* = 11; NCBI GEO profile GDS610 record) reduction in *Mlip* expression in the DMD affected patients while lamin A/C remained unaffected, suggesting that *Mlip* protein levels do not affect lamin A/C level of expression, whereas, shRNA mediated down-regulation of lamin A/C expression results in the up-regulation and cellular mislocalization of *Mlip* [5]. We previously showed that *Mlip* is localized to both the cytosol and nucleus of C2C12 myoblasts and is found within PML bodies that we believe is indicative of alternative functions beyond binding A-type lamins [5]. Since *Mlip* interacts with chromatin, then the shuttling of *Mlip* between these nuclear bodies, the nuclear envelope and cytosol, may be an important regulatory process that may be independent of its association with lamin A/C. Furthermore, this shuttling may be necessary for the coordinated withdrawal of myoblasts from the cell cycle and their fusion to form myotubes.

Not only does *Mlip* complex with chromatin, it also may function as a transcriptional regulator since *Mlip* is able to activate transcription. Moreover, this study reveals enrichment of putative *Mlip* targets with known involvement in a wide variety of developmental processes associated with the organogenesis of striated muscle and brain, suggesting an important role for *Mlip* in organ development. The reduction or absence of *Mlip* may have a non-specific global effect on the transcription of many genes, some of which may play an important role in muscle differentiation and function. *Mlip* may affect specific gene targets important for maintaining myogenic determination of skeletal muscle cells. This observation may have significance in the tissue-specific *Mlip* isoform distribution that was previously reported [5] in that these *Mlip* isoform patterns may dictate distinct transcriptional programs required for cell fate during development. This, in turn, may explain the phenotypic heterogeneity observed in laminopathy patients where over 400 mutations in the ubiquitously expressed LMNA gene have been described to date [41,42,43].

Previous studies have reported a critical role for lamin A/C and emerin in satellite cell differentiation [44,45,46] with mutations within lamin A of patients being attributed to Emery–Dreifuss and limb–girdle muscular dystrophy. The reduction of A-type lamin expression in C2C12 cells resulted in an up-regulation of *Mlip* expression and mislocalization of *Mlip* in the myoblast [5]. *Mlip*’s interaction with A-type lamins in the nuclear envelope may prove to be essential for normal differentiation. *MLIP* interacts with a domain found in the first 130 amino acids of A-type lamin [5]. Mutations, particularly those in the first 130 amino acids of lamin A/C, may alter the kinetics of the lamin A/C interaction with *Mlip* and result in reduced myotube formation. In C2C12 cells we observed the up-regulation of two additional *Mlip* splice forms as myoblasts differentiate and fuse to form myotubes. Since *Mlip* binds chromatin and also contains at least one transactivation domain, the loss or reduction of *Mlip* could affect the expression of numerous genes. The loss of *Mlip* has a functional consequence in mouse C2C12 myoblasts in that the temporal activation of both MyoD and MyoG were significantly reduced, which translated into a significant reduction in the fusion of these myoblasts to form myotubes. Because activation of MyoD and MyoG expression and subsequent differentiation was almost abolished in the absence of *Mlip*, we believe that in C2C12 myoblasts, *Mlip* is essential for muscle terminal differentiation and that it exerts its effects at least in part by enabling expression of myogenic regulatory factors. Our findings shed new light on the biological function of *Mlip* as a potential transcriptional co-factor and the role it may play in skeletal muscle differentiation.

## Figures and Tables

**Figure 1 cells-10-00615-f001:**
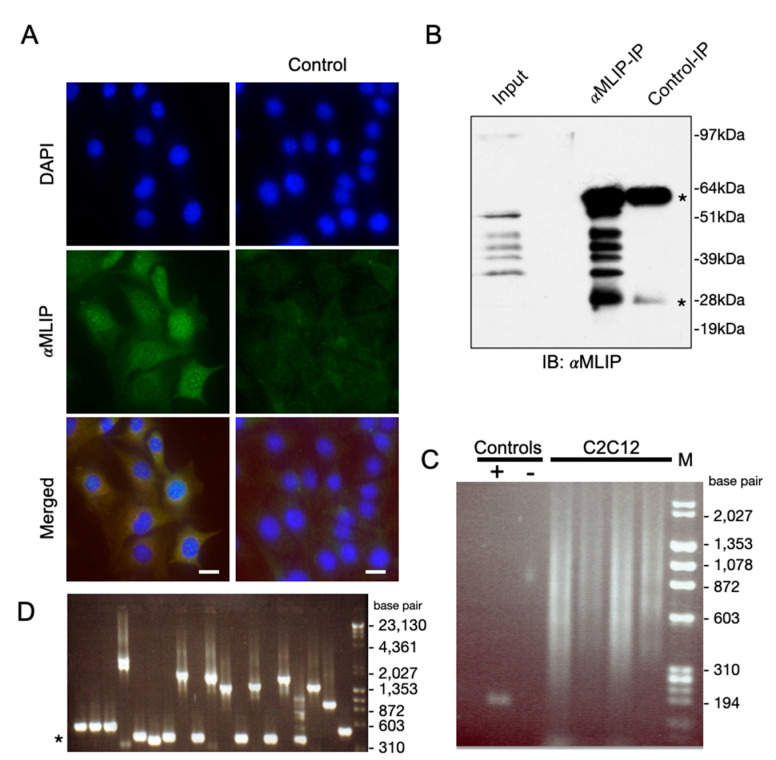
*Mlip* interacts with genomic DNA. (**A**) *Mlip* (green) is localized to the nucleus of C2C12 cell as discrete punctae. Chromatin is stained with DAPI (blue). Control: *Mlip*-shRNAmir mediated knockdown of endogenouse *Mlip* mRNA. 20× magnification; Scale bar = 10 µM (**B**) *Mlip* immunoprecipitations from HL1 whole cell lysates. The control lane used *Mlip*-peptide neutralized *Mlip* antibody. “*” IgG heavy and light chains. (**C**) Chromatin immunoprecipitation (ChIP) was performed with an *Mlip* specific antibody after chemical cross-linking. *Mlip*-ChIP enriched DNA (4 independent replicates) were isolated and resolved on an agarose gel and subsequently TA-cloned into the pCR-II plasmid and transformed into bacteria. Positive control (+), chromatin was immunoprecipitated with an anti-Histone 3 antibody; subsequently the immunoprecipitated chromatin was PCR amplified with primers targeting a known Histone 3 binding site. Negative control (−), normal rabbit pre-immune sera was used as an IgG negative control for the ChIP. (**D**) The *MLIP* mediated ChIP DNA was TA-cloned into pCRII. Direct PCR was performed off pCRII-*MLIP*-ChIP-DNA transformed colonies with primers targeting plasmid flanking regions of the *MLIP*-ChIP enriched DNA and visualized on an agarose gel. PCR product sequences were identified by DNA sequencing. “*” size of no insert cloned.

**Figure 2 cells-10-00615-f002:**
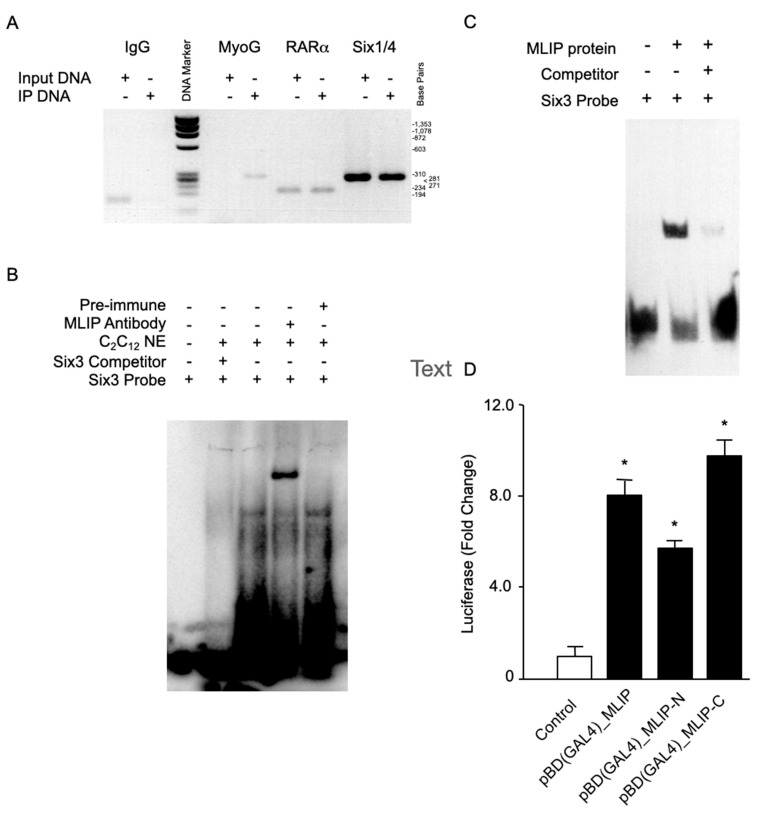
*Mlip* binds DNA and activates transcription. (**A**) Gene-specific PCR of *Mlip* genomic DNA targets. (**B**) Labeled SIX3 probe was mixed with 2 ug of C2C12 nuclear extracts (NE) and analyzed via gel shift assays. The complex was super-shifted by the addition of an *Mlip* specific polyclonal antibody. *Mlip* peptide neutralized antibody was unable to super-shift the SIX3 associated complex. (**C**) In vitro translated *Mlip* was tested for its ability to interact with labeled SIX3 probe by gel shift assay. The labeled SIX3 probe being competed away with a 50-fold molar excess of unlabeled SIX3 probe. (**D**) *Mlip* full-length, N-terminal (exons 1–3) and C-terminal *Mlip* (exons 5–12) fused to a GAL4 DNA binding domain trans-activates a GAL4-Luciferase promoter in HEK293 cells. The assay was repeated 3 times, * = *p* < 0.01 versus control, pBD-p53 (Agilent). ‘+’ = addition of reagent. ‘-’ symbol means ‘no reagent added’.

**Figure 3 cells-10-00615-f003:**
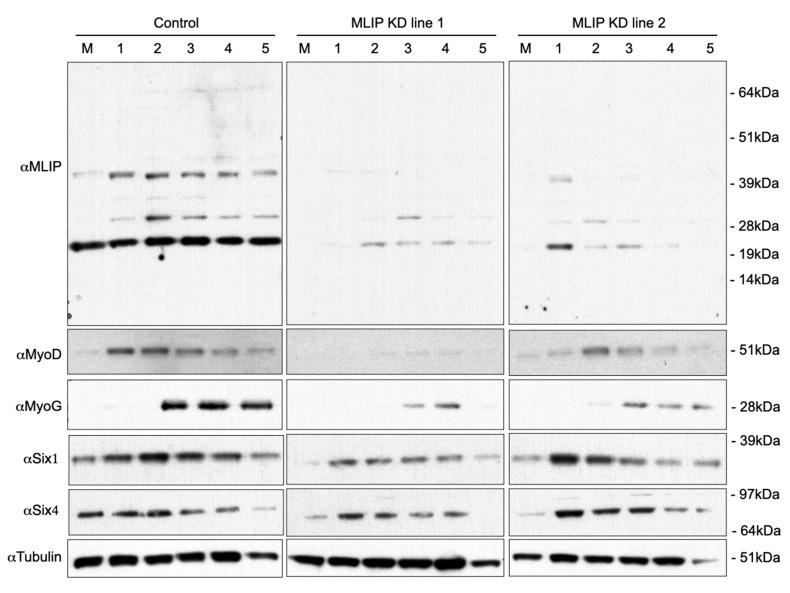
*Mlip* is necessary for C2C12 myoblast differentiation into myotubes. Western Blot analysis of C2C12 myoblasts (M) and C2C12 cells grown in differentiation media was performed using anti-*Mlip*, anti-MyoD, anti-MyoG, anti-Six1 and anti-SIX4 polyclonal antibodies. Lysates were isolated every day for 5 days of differentiation. Control cells were generated using a validated non-silencing negative control.

**Figure 4 cells-10-00615-f004:**
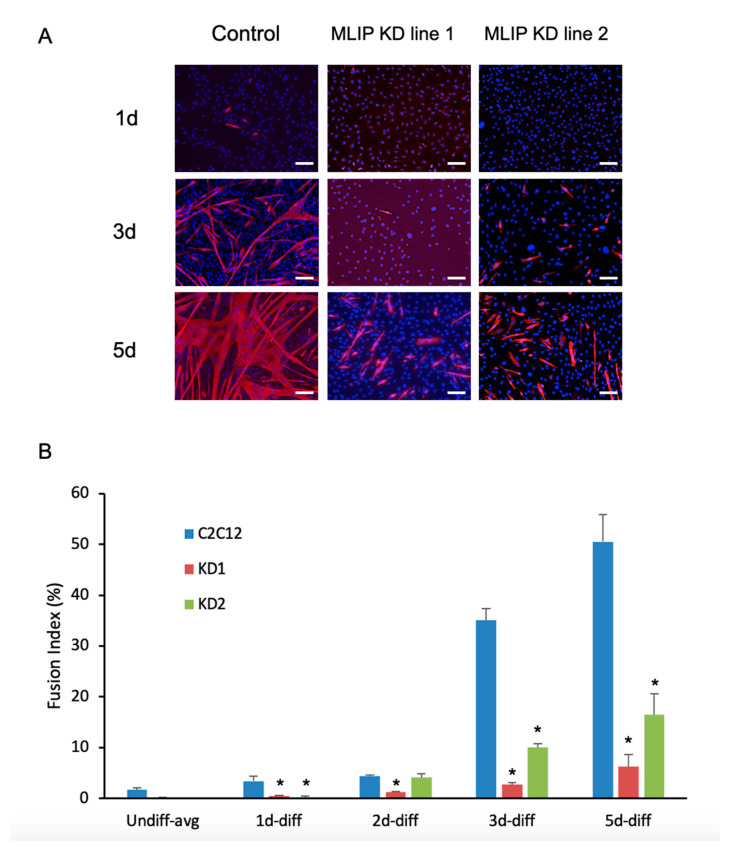
*Mlip* is required for normal myotube fusion and formation. (**A**) Indirect immunofluorescence of myosin heavy chain expression during C_2_C_12_ differentiation in the normal or in *MLIP*-depleted C2C12 myoblasts. Nuclei were stained with DAPI. 10 x magnification; Scale bar = 50 uM (**B**) Fusion index calculated as the fraction of total nuclei present inside myotubes (MyHC positive cells) after 0, 1, 2, 3 and 5 days of differentiation. The data represent the pooling of 3 independent differentiations with * = *p* < 0.001 vs. control. Control cells were generated using a validated non-silencing negative control.

**Table 1 cells-10-00615-t001:** Ontological analysis of cloned *Mlip* mediated chromatin immunoprecipitated DNA fragments.

Gene.	Differentiation	Commitment	Apoptosis	Proliferation	Survival	Growth
**Akt2**	**x**		**x**	**x**	**x**	**x**
**CREM**				**x**		
**SOX5**		**x**	**x**			
**PEL1**						
**KIF5C**						**x**
**PLC1**				**x**	**x**	**x**
**Met**				**x**	**x**	**x**
**MMP3**	**x**			**x**	**x**	
**Runx**	**x**		**x**	**x**		**x**
**Nek7**						
**FLI1**	**x**		**x**	**x**		**x**
**PP2R3A**					**x**	
**Notch2**	**x**	**x**	**x**	**x**	**x**	**x**

**Table 2 cells-10-00615-t002:** Biological processes enriched for on mouse chromosome 11 by *Mlip* mediated chromatin immunoprecipitation.

	*Mlip* Enriched Chr 11 Genes	All Chr 11 Genes
Gene Ontology Term (Biological Process)	*p*-Value	*p*-Value
0007275~multicellular organismal development	0.00004415	0.01820829
0032502~developmental process	0.00099698	0.03170131
0009987~cellular process	0.00146013	0.00000553
0016043~cellular component organization & biogenesis	0.00175770	0.00000001
0048731~system development	0.00243918	0.04057584
0006665~sphingolipid metabolic process	0.00257593	ND
0048856~anatomical structure development	0.00297968	0.02503862
0006886~intracellular protein transport	0.00359201	0.00028535
0048513~organ development	0.00386375	ND
0043407~negative regulation of MAP kinase activity	0.00476669	0.03816350
0009888~tissue development	0.00621520	ND
0006629~lipid metabolic process	0.00952032	ND

## Data Availability

Data sharing is not applicable to this article.

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
