# Peer review of "Muscle Enriched Lamin Interacting Protein (*Mlip*) Binds Chromatin and Is Required for Myoblast Differentiation"

_cells, 2021, doi:10.3390/cells10030615_

Round 1
Reviewer 1 Report
In this manuscript (cells-1062611), Ahmady and colleagues report MLIP’s interaction with chromatin and suggest its possible role as a transcriptional regulator. This work complements a previous study from the authors where they reported the identification of MLIP and its interaction and co-localization with Lamin A/C. Thus, this study provides an advance in the current knowledge of MLIP, particularly to the biological function of MLIP as a transcriptional co-factor and its possible role in skeletal muscle differentiation. Data presented support authors conclusions. However, the manuscript requires some modifications before being published.
Specific comments and suggestions include:
- Line 47. Reference Liu J, PNAS, 2020 should be added.
- Section 3.1 (lines 203-214) needs clarification. I understand that 133 putative MLIP chromatin targets were identified by ChIP-on-Chip (Table2) but it is not clear where the 45 putative MLIP gene targets from the “Analysis of the ontological enrichment of the 45 putative MLIP gene targets” come from. Did you select these 45 putative MLIP gene targets by DAVID bioinformatics resources analysis of the previous 133 selected targets? In addition, “a gene-specific ChIP-PCR was performed on a SELECTED number of identified MLIP targets to verify your ChIP-on-Chip data”. Which are the selected targets? Are they the ones shown in Figure 2A or are these only some of them? And how were they selected?
- In lines 227-233. “A mobility shift was observed for all of the targets in the presence of C2C12 nuclear extract.” For all of the targets? For the 133, the 45 or for the selected MLIP targets? And where is this data? Is this “data not shown”?
- In Section 3.3, when talking about differentiation time points in the text, you use hours but in Figure 3 you use days as time points. Please, do not mix time points, use only hours or only days.
- Line 275. “Upon differentiation of MLIP-KD1 and MLIP-KD2, we observed a DECREASE in MLIP expression…”
- Figure 1A and Figure 4A images lack the scale. Figure 2 legend lacks abbreviations.
Reviewer 2 Report
In their manuscript, Ahmady et al. report the ability of their recently discovered muscle-enriched A-type lamin-interacting protein (MLIP) to directly interact with chromatin and genomic DNA as well as to potentially function as a transcriptional cofactor. Using chromatin immunoprecipitation coupled with microarrays, the authors demonstrate that MLIP associates with genomic regions found in close proximity to genes important for tissue-specific cellular differentiation. They go on to show that C2C12 myoblasts depleted of MLIP exhibit a down-regulation of myogenic regulatory proteins and are unable to efficiently differentiate into myotubes in vitro. Overall, this is a generally well-written manuscript that addresses an exciting and timely topic in the field. However, I have several major and minor issues that I feel need to be dealt with prior to my recommending that it be accepted for publication. These issues are outlined below.
Major Issues:
- The authors state in the abstract, “MLIP may exert this role (i.e. myoblast differentiation) as a transcriptional regulator of a myogenic program that is unique only to amniotes and thus provides a new muscle signaling paradigm”. This is the major conclusion of the work presented in this manuscript, yet the only evidence that they provide to support this claim is that MLIP constructs fused to a GAL4 DNA binding domain trans-activates a GAL4-Luciferase promoter in HEK293 cells. While the authors do show that MLIP-depletion decreases the expression of key myogenic regulatory factors (Figure 3), this decrease could be due to mechanisms unrelated to the potential ability of MLIP to act as a transcriptional co-factor. In addition, it is unclear to me how MLIP represents a “new muscle signaling program”, as signal transduction was not explicitly examined in this work. Unless the authors can show that MLIP-depletion impacts the transcription of genes important for myoblast differentiation (e.g. via RNAseq experiments or northern blotting), I would strongly recommend that they tone down their conclusion accordingly.
- Lines 104-105: it is insufficient to simply state, “Images were then visualized using fluorescent microscopy (Carl Zeiss Microscope). They need to provide the information listed below:
- Exactly which Zeiss microscope was used and which modality (e.g. epifluorescence, confocal, etc.).
- The magnification, correction level (e.g. PLAN-APO), and numerical aperture of the objectives used.
- The light source used.
- The camera used.
- The fluorescence excitation and emission filters used.
- The software used to take the images. I assume that it was Zeiss’ Zen software, but this could be different.
- Line 182: the authors need to explicitly explain how they performed their “Ontological analyses”. Without this information, it is difficult for their readers to know how to reproduce their findings.
- Scale bars need to be provided in at least one of the images shown in Figures 1A and 4A.
- Molecular weight standard markers need to be labeled with the appropriate sizes in all immunoblots and agarose gels.
- Based on the images shown in Figure 1A, I am very worried about the specificity of the anti-MLIP antibody that the authors are using in this study. It appears to interact non-specifically with proteins in the cytoplasm as well as with proteins within the nucleoplasm. While I agree that the amount of nucleoplasmic punctae seems to be decreased in the control images (Figure 1A), in the immunoblot of the control IP (Figure 1B), and in the immunoblots of the MLIP KD lines (Figure 3), I am hoping that the authors can find a better set of images for Figure 1A. In addition, it would be good to see the entire αMLIP immunoblots shown in Figure 3, not just a small section of them. This information would help the readers better evaluate the data presented in this manuscript.
- The size of the expected control band in Figure 1C needs to be indicated. Otherwise, the readers do not know how to interpret the blot presented here. Also, it is unclear what each of the 4 C2C12 lanes represents. Are these just biological replicates?
- Based on the legend for Figure 1D, it is unclear what exactly is being shown here. The authors need to more explicitly label their gels; otherwise, their readers will not know what they are looking at.
- Line 258: The title of section 3.3 is “Impaired myotube formation in the absence of MLIP”. I strongly disagree with the use of the word “absence” here (and throughout the manuscript), as the authors only use shRNA-mediated knockdown to interrogate the requirement of MLIP for myotube formation. Thus, I recommend that the authors swap “the absence of MLIP” for “in MLIP-depleted C2C12 myoblasts”.
- I would very much like to know if the authors can rescue the phenotypes that they observe in their MLIP-depleted C2C12 cells by re-expressing a cDNA construct that encodes a shRNA-resistant version of MLIP. This is the gold standard in the field for determining if a phenotype caused by RNAi-mediated depletion occurs specifically due to the depletion of the intended protein of interest and not due to off-target effects. I realize that the authors used a mix of 3 different shRNA sequences to generate their MLIP-KD cell lines; however, this is not enough to rule out the possibility that the phenotypes observed in these cells are not due to off-target effects. Being able to rescue the phenotypes caused by MLIP-depletion in their assay would enable the authors to perform powerful structure-function-based experiments designed to investigate the mechanism underlying the contribution of MLIP to myoblast differentiation.
- The authors should quantify and perform statistical analyses of the results of their Western blots shown in Figure 3 so that they can discuss these results in terms of “significance”.
- In lines 354-356, the authors postulate that the shuttling of MLIP between “these nuclear bodies, the nuclear envelope and cytosol” might be necessary for the coordinated withdrawal of myoblasts from the cell cycle and their fusion to form myotubes”. Since the author show that the depletion of MLIP from myoblasts impairs their fusion index following their transfer into differentiation media, do the authors know if MLIP-depletion negatively impact the growth and/or cycling of C2C12 cells? This information would help the authors better determine the mechanism underlying the contribution of MLIP to myogenesis.
- Does the depletion of MLIP influence the expression, localization, and/or stability of A-type lamins in myoblasts and/or myotubes? Since the authors know that depleting A-type lamins “results in the up-regulation and mislocalization of MLIP”, it would be mechanistically good to know if MLIP-depletion impacts A-type lamins in any way.
- Does the localization of MLIP differ between myoblasts and myotubes?
- What happens if the authors allow the MLIP-depleted C2C12 myoblasts to continue differentiating for 8 days? Do they catch up with the control C2C12 myoblasts? It would also be good if the authors could provide a plot of fusion indices for their control and MLIP-depleted C2C12 myotubes at 3, 5, and 8 days post differentiation. Furthermore, the authors should provide an additional plot of the average number of nuclei per myotube at 3, 5, and 8 days post differentiation. This information is standard for these types of experiments and it would allow the authors to better characterize the effect of MLIP-depletion on myoblast differentiation.
- Given that MLIP interacts with A-type lamins, is there any relationship between the genomic regions that the authors showed can associate with MLIP and the so-called lamina-associated domains in C2C12 myoblasts? In other words, is it possible that MLIP acts during myoblast differentiation as a tether that controls the association of genomic regions important for this process with the nuclear lamina present at the nuclear periphery? It seems that this may be an alternative model for MLIP function during myoblast differentiation that could be addressed in the manuscript by the authors.
Minor Issues:
- The authors should refrain from using contractions in their manuscript (e.g. “don’t”, line 347).
- Please define all abbreviations the first time that they are used in the manuscript. Once an abbreviation is used, there is no need to redefine it again (e.g. DMD, line 341).
- I would recommend that the authors insert a “the” in the following places:
- Line 337: before “diaphragm”.
- Line 338: before “heart”.
- Line 360: before “organogenesis”.
- Human gene names are capitalized and written in italics. Mouse gene names are written in italics with the first letter capitalized. This needs to be addressed throughout the manuscript.
- The authors need to provide the dilutions that they used for their primary and secondary antibodies somewhere in the Materials and Methods section.
- The authors should limit the amount of passive voice that they use in their writing.
- There are several errors in subject/verb agreement throughout the manuscript that need to be addressed. For example:
- Line 309: “mutation” and “translates” need to be changed to “mutations” and “translate”, respectively.
- Line 311: “cell” needs to be replaced with “cells”.
- Line 337: “muscle” should be “muscles”.
- What is the red blob present in upper left corner of the images shown in Figure 1A?
- Line 181: the authors should provide a reference for TA cloning here, as not everyone may be familiar with this method.
- Lines 37-38: the authors mention that the MLIP gene “encodes at least seven, alternatively spliced, LMNA-interacting factors that possess several structural motifs not found in other proteins”. It would be best if the authors could replace “LMNA” with “A-type lamin”, as I assume that they mean to say that MLIP interacts with the proteins encoded by LMNA. Also, it would be helpful for their readers if the authors could elaborate on the “several structural motifs”.
- Line 43: the authors should elaborate on what they mean by “mislocalization of MLIP” here.
- Line 44: the authors first mention IsI1 here, but they fail to explain what this protein is or what it does. The reader has to wait until lines 329-330 for this information.
- Lines 211-214: The authors state, “We observed an enrichment of the myogenin (MyoG), retinoic acid receptor alpha (RARα) and Sine oculis homeobox homolog 1 (Six1/4) in MLIP mediated immunoprecipitated chromatin over the whole cell extract control (Figure 2A)”. However, I only see that MyoG is enriched. Perhaps the authors could better explain their experiment here?
- What do the “*” represent in Figures 2C-B and the “**” in Figure 2B? This needs to be explained in the figure legend somewhere.
- I feel like the information shown in Table 2 should be moved to supplemental material.
- Line 261: the authors should provide a reference for shRNAmir technology here, as not everyone may be familiar with these constructs.
- Line 309: the authors wait until the Discussion to mention “laminopathies”. This word should be used somewhere in lines 27-30, when laminopathies are discussed for the first time in the Introduction.
- Line 311: the authors need to provide a reference for the statement, “an established in vitro myoblast model system”.
- Line 369-370: the authors need to provide a reference for the statement, “over 400 mutations in the ubiquitously expressed LMNA (sic italics) gene have been described to date”.
- Line 372: the references 34-35 made in this line are not provided in the References section of the manuscript.
- Lines 376-377: the authors state, “Mutations, particularly those in the first 130 amino acids of lamin A/C, may alter the kinetics of the lamin A/C”. However, it is unclear why they are making this statement. Does MLIP interact with lamin A/C within its first 130 amino acids? It is also unclear what “the kinetics of the lamin A/C” are. I assume that the authors mean “polymerization kinetics”, but I do not want to guess.
Reviewer 3 Report
This is an interesting study examining the role of MLIP in muscle cell differentiation. MLIP is a lamin A binding protein that also interacts with chromatin. The authors in this paper do a nice job understanding which regions of DNA this protein binds to in muscle cells.
An additional positive is the authors showing that MLIP is important in muscle cell fusion, and this finding is quite compelling that this protein is significant in muscle. My only question is which interaction is more important, the MLIP to lamin A, MLIP to DNA or bridging the lamina to the DNA. I think the authors make a compelling case that the MLIP to DNA is important, but given how much is know about the importance of lamin A, I wonder if instead this protein could just be altering lamin A expression or nuclear morphology. Did the authors investigate this in the KD cells? For example, I see very large nuclei in some of the representative images in Figure 4, is this significant?
Round 2
Reviewer 2 Report
I have attached a PDF containing my comments.
